# Response Surface Modeling of the Steady-State Impedance Responses of Gas Sensor Arrays Comprising Functionalized Carbon Nanotubes to Detect Ozone and Nitrogen Dioxide

**DOI:** 10.3390/s23208447

**Published:** 2023-10-13

**Authors:** Krishna Naishadham, Gautam Naishadham, Nelson Cabrera, Elena Bekyarova

**Affiliations:** 1Wi-Sense LLC., Atlanta, GA 30082, USA; gnaisha2@gmail.com; 2Carbon Solutions, Inc., Riverside, CA 92507, USA; sales@carbonsolution.com (N.C.); bekyarova@carbonsolution.com (E.B.)

**Keywords:** carbon nanotubes, environmental pollution, gas sensor, impedance, nanotechnology, ozone, nitrogen dioxide, personal exposure, low-cost sensors, response surface

## Abstract

Carbon nanotube (CNT) sensors provide a versatile chemical platform for ambient monitoring of ozone (O_3_) and nitrogen dioxide (NO_2_), two important airborne pollutants known to cause acute respiratory and cardiovascular health problems. CNTs have shown great potential for use as sensing layers due to their unique properties, including high surface to volume ratio, numerous active sites and crystal facets with high surface reactivity, and high thermal and electrical conductivity. With operational advantages such as compactness, low-power operation, and easy integration with electronics devices, nanotechnology is expected to have a significant impact on portable low-cost environmental sensors. Enhanced sensitivity is feasible by functionalizing the CNTs with polymers, metals, and metal oxides. This paper focuses on the design and performance of a two-element array of O_3_ and NO_2_ sensors comprising single-walled CNTs functionalized by covalent modification with organic functional groups. Unlike the conventional chemiresistor in which the change in DC resistance across the sensor terminals is measured, we characterize the sensor array response by measuring both the magnitude and phase of the AC impedance. Multivariate response provides higher degrees of freedom in sensor array data processing. The complex impedance of each sensor is measured at 5 kHz in a controlled gas-flow chamber using gas mixtures with O_3_ in the 60–120 ppb range and NO_2_ between 20 and 80 ppb. The measured data reveal response change in the 26–36% range for the O_3_ sensor and 5–31% for the NO_2_ sensor. Multivariate optimization is used to fit the laboratory measurements to a response surface mathematical model, from which sensitivity and selectivity are calculated. The ozone sensor exhibits high sensitivity (e.g., 5 to 6 MΩ/ppb for the impedance magnitude) and high selectivity (0.8 to 0.9) for interferent (NO_2_) levels below 30 ppb. However, the NO_2_ sensor is not selective.

## 1. Introduction

Traffic emissions, industrial pollution, climate change-related wildfires, and indoor sources, such as fossil fuel-burning combustion appliances, cigarette smoke, and building materials, release harmful levels of ozone (O_3_), nitrogen dioxide (NO_2_), volatile organic compounds (VOCs), and particulate matter (PM) into the environment [1,2,3,4,5,6,7,8,9,10,11,12,13,14,15,16,17]. VOCs and NO_2_ are also precursors of secondary ozone emissions [8] in a temperature-dependent photochemical process. The health effects of air pollution exposure are widespread, with negative implications for the cardiovascular, respiratory, immune, and nervous systems [18,19,20,21,22,23,24,25,26]. The global burden of disease attributable to ambient air pollution is at a historical high, with over 6.7 million deaths per year [27,28]. Anthropogenic climate change caused by global warming, with temperatures anticipated to increase by 1.5–2 °C by the end of this century [1], is likely to worsen mortality and morbidity, especially in cardiovascular and respiratory diseases [18,19,20,21,22,23,24]. In the United States alone, air pollution emissions are estimated to cause 200,000 premature deaths per year, with 55,000 of those attributable to transportation-related emissions [29]. 

Scientific studies on air pollution and its health effects indicate that the assessment of personal exposure to environmental pollutants using mobile portable sensors can improve public health and occupational health [4,5,6]. It is very important to characterize the inhaled levels within the breathing zone, and in real time, because O_3_ and NO_2_ display substantial spatial or temporal heterogeneity [8,9]. The regulatory air quality monitors are sparsely located (mostly in urban areas) and thus measure pollutant levels, which may differ substantially from those measured near the subject [4]. The resolution of exposure to harmful pollutants is severely limited by the distance from the monitoring stations and the transportation models used for extrapolation to residential neighborhoods [5,6]. Secondly, sample averaging (e.g., 8-h averages are reported for O_3_ [8]) is known to miss transient high-level exposures [30], and could increase the risk of airway inflammation in respiratory diseases [21]. A low-cost, personal monitor will address this void in fine-scale exposure data and can potentially benefit patients who suffer from allergies and air pollution, such as those with asthma and chronic obstructive pulmonary disease (COPD). The desired characteristics of a personal monitor include small size and weight, portability or wearability, low power consumption, battery operation, wireless connectivity, and linear response with high sensitivity to environmental gases in the parts per billion (ppb) range (e.g., 20 to 100 ppb for O_3_). 

The operation principle of gas sensors involves measuring changes in the specific properties of a sensing material (e.g., resistance, capacitance, mass, current, etc.) upon exposure to a gas species. These changes can be measured either directly or indirectly. A typical gas sensor consists of a sensing layer, deposited on a transducing platform that is in contact with the environment, together with an electronic circuit that produces a measurable output signal. The performance of a gas sensor is evaluated by considering several indicators: sensitivity, selectivity, response time, stability, power consumption, and reversibility. The predominant gas sensing technologies are conductimetric, capacitive, gravimetric (e.g., quartz crystal microbalance), surface acoustic wave resonators, colorimetric, optical spectroscopy, UV absorbance, photoionization detection, electrochemical (EC), heated metal oxide semiconductors (HMOS), and carbon nanotubes (CNTs) [31,32,33,34]. Of these methods, low-cost commercial gas sensors for environmental monitoring [35,36,37,38] mostly employ EC [39,40,41,42,43,44,45,46,47] and HMOS [48,49,50,51,52,53,54,55,56,57,58,59,60,61,62,63] methods. 

Nanostructured materials have shown great potential for use as sensing layers due to their unique properties, including high surface to volume ratio, numerous active sites and crystal facets with high surface reactivity, and high thermal and electrical conductivity [64,65]. With operational advantages such as enhanced sensitivity, low-power operation, and easy integration with electronics devices, nanotechnology is expected to have a significant impact on portable, inexpensive gas sensors. Unlike HMOS sensors, nanomaterial-based sensors operate at low temperatures (below 150 °C). At present, room temperature operation is feasible for gas detection only at high concentrations [65]. 

Iijima’s discovery of CNTs in 1991 [66] played a significant role in the evolution of nanomaterials, including nanoparticles (NPs), zeolites, nanorods, and graphene. Advances in nanotechnology have led to prolific applications in sensing, communication, and electronics spanning consumer, military, industrial, and medical use [67,68,69,70,71,72,73,74,75,76,77,78,79,80,81,82,83]. CNTs have been used by several researchers to demonstrate laboratory detectors for environmental gases, such as ammonia, nitrogen, NO_2_, CO_2_, etc. [67,68,69,70,71,72], as well as deadly gases, such as the nerve agent sarin [73,74,75]. These *chemiresistor* sensors measure the change in DC resistance between a pair of electrodes resulting from the interaction of the CNTs with the surrounding environment [69,70,71,72,73,74,75]. Low-frequency AC phenomena based on near-field inductive coupling [76,77,78,79], and the *chemicapacitor* [80,81,82], in which the sensing mechanism is based upon changes in the effective permittivity of the CNT layer deposited on a dielectric substrate, have been shown to improve sensitivity in comparison with chemiresistors. Optical methods, such as plasmonic optical transmission for ammonia detection [83] and laser scattering for particulate matter (PM) detection [84], have the potential to improve accuracy and stability in environmental sensing. However, miniaturization of optical sensor systems to enable wearability is challenging, and commercially available PM sensors are predominantly handheld. By integrating the CNT detector with microwave antennas, nanotechnology has the potential to be used in wireless sensor nodes deployed at remote sites to monitor environmental pollutants and characterize human health exposure. Gas sensors coupled to microwave antennas, such as patch and dipole resonators, have been introduced, demonstrating resonance shift discrimination as an accurate means of ammonia gas detection [85,86,87,88,89]. Recently, microwave resonance-based gas discrimination has been improved to isolate the antenna from the sensor, resulting in enhanced performance of a humidity sensor [90,91,92].

In order to expand the impact of nanomaterial-based sensors beyond the laboratory discovery stage into the commercial arena for environmental sensing, it is necessary to improve their sensing performance in a multi-analyte environment. In nanomaterials such as CNTs, the sensor response to a target gas can be tailored for improved specificity by functionalizing the materials with dopants such as metal oxides, metals, and conductive polymers. Chemical functionalization enhances both processability and sensing performance [69,93,94,95,96,97]. First, it allows the unique properties of CNTs to be enhanced by binding them with dopants to create hybrid sensing materials with enhanced sensitivity and a faster response time. Second, it can improve the dispersion of CNTs in various solvents, including water, which enables cost-effective, scalable methods to manufacture sensors by simple dispensing or printing techniques. Third, surface modification of CNTs by attaching functional groups makes it possible to tune their interaction with the surrounding environment and potentially enhance the selectivity of gas sensors.

Cross-sensitivity to other oxidizing gases remains a major problem in the detection of environmental pollutants, such as O_3_ and NO_2_, wherein low levels (below 100 ppb) need to be measured. Ozone and nitrogen dioxide are oxidizing gases with similar chemical affinity and produce nonspecific responses in EC [42,43,44,45,46], HMOS [48,49,50,51,57,58,59], as well as nanotechnology-based [97] sensors. Cross-sensitivity to the interfering gas decreases the selectivity to the target gas. A commercial EC gas sensor utilizes membrane filters in the ozone sensor to trap NO_2_ (and vice versa) [38]. This filter has a life span of a few ppm hours (product of exposure concentration and the total exposure time in hours), requiring frequent replacement and adding to the operational cost. In MOS sensors, selectivity can be improved by temperature modulation of the heater current [53,60,61], sometimes in an array combination with multiple sensing elements [62,63]. 

In this paper, the design and performance of a sensor array comprising sensor films with single-walled nanotubes (SWNTs) for the detection of O_3_ and NO_2_ is presented. Pristine SWNTs are functionalized by covalent modification with octadecylamine (ODA) groups [97] for O_3_ sensors. Amide (CONH_2_) functionalized SWNTs doped with zinc oxide (ZnO) nanoparticles are utilized for NO_2_ sensors. The amide-functionalized SWNTs are prepared by reacting carboxylated nanotubes with ammonia. The amide group is a much stronger base than carboxylic acid, but not as basic as amines. Thus, the amide-SWNTs are designed to have higher adsorption affinity for NO_2_, i.e., a stronger response, and because they do not bind to the analyte molecules strongly, these SWNTs tend to have faster sensor recovery. The pristine SWNTs are synthesized using the arc-discharge method, which produces a mixture of metallic and semiconducting SWNTs with very high crystallinity [96]. These nanomaterials are dispersed in a compatible solvent using ultrasonication to break the agglomerates and prepare the ink for sensor deposition. The ink is then spray-coated on interdigital electrodes fabricated on a fiber-reinforced glass epoxy substrate (FR-4). An array comprising O_3_ and NO_2_ sensors (one each) is fabricated and tested in the laboratory with the two gases. The fabricated sensor array printed circuit board (PCB) is small (25 × 25 × 0.8 mm^3^), and produced using standard photolithography (OSH Park, Portland, OR, USA). The array is calibrated using a mixture of O_3_ and NO_2_ gases. The sensor performance is described by deducing the sensitivity, selectivity, and response times from the calibration measurements. The calibration standards, to which the gas sensors are calibrated, are benchtop reference instruments that measure the precise levels of O_3_ (Model 202 Ozone Monitor, 2B Technologies, Broomfield, CO, USA) and NO_2_ (Model 405 nm Chemiluminescence NO_x_ Monitor, 2B Technologies, Broomfield, CO, USA) in the controlled-flow gas chamber containing the sensor array. 

Two innovations are proposed in this paper. The first innovation involves stimulation of the sensor with an AC signal in a narrow frequency band between 1 and 10 kHz and measuring dynamic changes in the sensor impedance (magnitude and phase). A preliminary design of this impedance measurement circuit (IMC) for a single sensor element is presented in [98]. In this paper, this design is extended to a 2 × 2 sensor array for measuring the multiplexed magnitude and phase sensor responses to each gas. In contrast with chemiresistor measurements, which involve static measurements of DC resistance, arraying multiple sensors with magnitude and phase responses enriches signal diversity (i.e., two independent response features for each sensor) and improves the signal-to-noise ratio [99]. In the second innovation, a multivariate nonlinear regression algorithm based on the response surface method (RSM) [100] is applied to the gas sensor calibration. RSM calculates the calibration factors, including the interaction term, to account for cross-sensitivities among the nonselective sensor responses to (O_3_, NO_2_) gas mixtures. The response surface mathematical model is utilized to evaluate the sensor performance, including sensitivity and selectivity.

A combination of controlled laboratory testing, field measurements in outdoor ambient air, and reference data from nearby air quality monitors can be used to augment the RSM regression model with supervised training and validation based on machine learning (neural networks, support vector machine, artificial intelligence, etc. [101,102,103,104,105,106,107,108,109]). Besides improved accuracy and localization of real-time measurements in ambient environments, these machine learning approaches enable real-time self-calibration. However, this paper focuses on demonstrating the RSM using only laboratory calibration measurements. The outdoor ambient measurements and machine learning models are the subject of ongoing research.

The paper is organized as follows. Section 2 describes the functional CNT materials used in the sensor array, the design of the sensor array and the electronics PCB, and the pertinent laboratory calibration experiments. The data analysis using the RSM fitting of the measured sensor responses is presented in Section 3. The sensor performance, including sensitivity and selectivity, is also discussed in Section 3. A brief discussion of the analyzed results and the potential for future research on improvements in sensor array performance are presented in Section 4.

## 2. Materials and Methods

### 2.1. Nanomaterial Synthesis and Deposition

Our objective is to functionalize the electronic nanostructure of pristine SWNTs to improve sensor performance in the detection of strong oxidizing gases such as O_3_ and NO_2_. The inks of the functional nanomaterial are deposited on the sensing electrodes. The SWNT materials are dispersed in a solvent by ultrasonication in a bath sonicator (Aquasonic HT50) for about 1 h to obtain a well-dispersed stock, which is diluted before deposition on electrodes. Printed circuit boards are designed and fabricated on commercially available FR-4 substrates with interdigitated electrodes (IDEs) laid out on top of the board. The functionalized SWNT ink is spray-coated on these electrodes using a commercial-grade air-brush spray coating apparatus (Master Airbrush). The DC resistance of the sensor is measured in situ during the deposition. The substrate is sprayed intermittently until the desired resistance is reached, and then annealed at 80 °C to 100 °C for 1–2 h to remove residual solvent. The final resistance is measured after overnight equilibration in air. The devices are inspected under a microscope to ensure uniformity and lack of contamination. 

#### 2.1.1. Ozone Sensors 

Ozone sensors comprising pristine SWNTs functionalized by covalent modification with octadecylamine (ODA) groups were prepared using the SWNT functionalization concepts described in [93,94]. The base SWNTs (P3-SWNT available from Carbon Solutions, Inc. (CSI)) were purified with nitric acid, which introduces 1.0–3.0 atomic % carboxylic acid groups on the nanotube walls, and can be derivatized with a variety of functional groups [110]. The carboxylic acid groups are used to covalently attach ODA molecules to SWNT walls [94]. These organic molecules provide a medium for the adsorption of ozone. The ODA-SWNTs are dispersed in tetrahydrofuran (THF), in which they have the highest dispersibility [94], using ultrasonication to break the bundles and prepare the ink for sensor deposition. 

The functionalized ODA-SWNT nanomaterial is characterized for its properties, and Figure 1 displays the results. Scanning electron microscopy (SEM) indicates the morphology of the ODA film as the formation of a highly porous network of fibrous CNT bundles decorated by the ODA molecules (Figure 1a). Thermogravimetric analysis (TGA) of the functionalized materials determines mass losses and presents information on their thermal stability. TGA was performed in air at a heating rate of 5 °C min^−1^ using a Perkin Elmer Pyris 1 apparatus. Figure 1b indicates that the nanomaterial is stable until about 200 °C, and there is no indication of any significant thermal decomposition below 250 °C. The metal content from the catalysts Ni and Y is 4 weight%, as estimated from the TGA residue. Comparison to the base SWNT material, which contains 5.2 weight% metal, gives an estimated ODA to SWNT weight ratio of 1:3.

The mid-IR spectrum, shown in Figure 1c, probes the functional groups present in the material and provides evidence of covalent functionalization. Upon functionalization with alkyl amines, we observe bands around 2922–2853 cm^−1^, which correspond to the C-H stretching vibrations of the alkyl chain (CH_2_). The C-H bending vibration is observed at 1463 cm^−1^. The covalent attachment of the ODA molecules to the SWNTs is verified by the detection of the typical C=O stretching vibration at 1651 cm^−1^. The near-IR spectrum in Figure 1d illustrates the second semiconducting inter-band transition (S_22_) of the SWNTs, which indicates the preserved electronic structure of the carbon nanotubes after chemical functionalization.

#### 2.1.2. Nitrogen Dioxide Sensors 

A composite of SWNTs functionalized with amide groups (CONH_2_), and doped with zinc oxide (ZnO) nanoparticles, is used as the nanomaterial for NO_2_ sensing. This material, denoted as Zn-Amide-SWNT, has the dual advantages of SWNTs and metal oxides for improving sensor performance. The amide-functionalized SWNTs were prepared by reacting carboxylic acid-functionalized SWNTs with ammonia. The nanomaterial is stable in air till 200 °C, as shown by the TGA in Figure 2a. The presence of amide groups is verified by the mid-IR spectrum in Figure 2b, which displays the stretching vibration of C=O at 1640 cm^−1^ and N-H vibration at 3400 cm^−1^. The near-IR spectrum in Figure 2c illustrates the preserved electronic structure of the SWNTs with a well-defined S_22_ transition.

The amide group is a much stronger base than carboxylic acid, but not as basic as amines. ZnO is an n-type semiconducting metal oxide, with electrons as the main carriers. Different forms of ZnO have been utilized for the preparation of metal oxide sensors for NO_2_ detection [111]. The adsorption of NO_2_ on the surface of ZnO nanoparticles creates an electron-depleted layer and alters the electrical properties of the material to produce a sensitive response. This property makes ZnO very attractive for NO_2_ detection in conventional HMOS sensors. The drawbacks of these HMOS sensors are high operating temperature (above 300 °C) and low electrical conductivity (due to electron depletion). However, metal oxide semiconductors (SnO_2_, ZnO and WO_3_) have the advantages of easy fabrication, high sensitivity, and low detection limit. Metal oxides, in general, are good sensing materials, but they are poor conductors or transducers of the electrical signal. A hybrid sensor material that combines the sensing capability of a metal oxide with the exceptional electronic conductivity of SWNTs is expected to improve sensor performance [111]. The amide-functionalized SWNT-ZnO sensors will operate at a much lower temperature (in comparison with pure ZnO) because of the fast charge transfer afforded by the nanotubes. A recent report in the literature on ZnO-SWNT sensors confirms the viability of this approach [112].

For the synthesis of Zn-amide hybrid nanomaterial, we dispersed amide-SWNTs in the solvent dimethyformamide (DMF) and added ZnO nanoparticles by isolating the NPs from ZnO Nanoshield ZN-5060. Analysis of the particle size was not performed, but the expected average size is 50–100 nm. The mixture was sonicated and then refluxed overnight, followed by additional sonication before deposition. The prepared ink has a composition of ZnO to SWNTs in the ratio 1:1 to 1:2. A two-step process was used for the deposition of this nanomaterial. The Zn-amide SWNT sensors were prepared by first spraying a thin film of amide-SWNTs (using DMF dispersion) on the electrodes and then depositing the ZnO-SWNTs by drop-casting in order to assure the exact material composition of the sensor films desired.

### 2.2. Sensor and Electronics Board Design

#### 2.2.1. Sensor Array 

The 2 × 2 array layout on the FR-4 substrate is shown in Figure 3. Each sensor comprises a pair of IDEs laid out on one side of the sensor PCB, and a meandered line heater element printed on the other side. All heater elements are connected in parallel. The purpose of the low-temperature (between 75 and 100 °C) heater is to accelerate the desorption of the gas from the activated CNT surface over the electrodes. The four pairs of terminals on the top and bottom of the layout in Figure 3 connect the sensors to the impedance measurement circuit, to be described shortly. The terminals on the left connect the heater to a 3.3 V power supply, and those on the right measure the resistance of a thermistor (not shown) mounted between the two narrow inner pads shown.

A slightly modified version of the layout in Figure 3 has been fabricated on a 1.6 mm thick FR-4 substrate. In order to miniaturize the footprint, the heater pads were routed to the top, and the thermistor pads for monitoring the heater temperature to the bottom, so that we have two parallel rows with six pads in each row (see Figure 4). For the purposes of this paper, we did not use the sensor positions S3 and S4, which were uncoated. Position S1 had a coating of ODA-SWNTs for O_3_ detection, and position S2, a coating of Zn-Amide-SWNTs for NO_2_ detection. The header pins shown are used to connect the terminals of the sensors, the thermistor and the heater, to the impedance measurement pads on the electronics board. A 3.3 V DC supply powers the heater and the electronics.

#### 2.2.2. Impedance Measurement Circuit 

The CNT thin films deposited on the IDEs produce an electronic signal upon interaction with the analyte. This transduced signal is proportional to the vector impedance (magnitude and phase) between the electrodes, measured using the Analog Devices AD5933 Low Frequency Impedance Converter chip [113]. We designed the impedance measurement circuit (IMC) on an FR-4 PCB for the integration of AD5933 with a low-power microcontroller (Atmel ATmega328P, Microchip technology, Inc., Chandler, AZ, USA), 8-channel input and output analog multiplexers (Texas Instruments CD4051), a voltage regulator (LP5907, Texas Instruments, Dallas, TX, USA), and the signal conditioning circuitry to acquire, process, and record the sensor signals from a 2 × 2 sensor array. The laboratory prototype of this electronics board, designed using breakout boards for the Arduino microcontroller circuit (Adafruit Pro Trinket 5 V, 16 MHz, Adafruit, NY, USA) and a USB-to-serial adapter (FTDI FT232RL, Future Technology Devices International Limited, Glasgow, UK), is shown in Figure 5. The USB adapter converts the I2C signal output from the microcontroller to the USB so that a Raspberry Pi or a laptop computer can log the array measurements continuously. 

### 2.3. Sensor Calibration 

Wi-Sense designed and built a monolithic Teflon chamber, as shown in Figure 6, facilitating controlled gas flow to calibrate the sensors using either a single gas or a mixture of gases. As needed, we can efficiently test up to five sensor arrays in parallel using a single gas source with an appropriate flow rate. In this investigation, we calibrated the sensor array shown in Figure 4 for O_3_ and NO_2_. The gas chamber is fitted internally with a precision commercial humidity sensor to continuously monitor humidity levels. The thermistor on the sensor array board measures the temperature inside the chamber. The chamber has a rectangular window where the sensor array, mounted on a rigid board, is pushed snugly into the chamber wall. Thus, the inside face of the rigid board has sensors exposed to the gas, while the outside face has sensor leads connected by wires to the IMC electronics board shown in Figure 5. This chamber has been designed for seamless connection to electronic test equipment such as controlled voltage or current sources, function generators, impedance measurement equipment, and digital multimeters. This facilitates the calibration of several multiplexed sensors simultaneously using gas mixtures.

The 2B Technologies Model 714 NO_2_/NO/O_3_ Calibration Source produces O_3_ by photolysis of oxygen in scrubbed ambient laboratory air, and calibrated concentrations of NO are produced via photolysis of nitrous oxide (https://www.twobtech.com/model-714-no2noo3-calibration-source.html (accessed on 10 December 2022). Nitrous oxide is provided by disposable 8 or 16 oz cartridges typically used for making whipped cream, eliminating the requirement for a compressed gas cylinder and thereby enhancing safety and portability. Calibrated concentrations of NO_2_ are produced by gas phase titration of NO with O_3_. The 2B Technologies Model 202 Ozone Monitor provides accurate and precise reference measurements of ozone ranging from low ppb (precision of ~1.5 ppb) up to 250 ppm based on the well-established technique of absorption of UV light at 254 nm wavelength. The measurement setup shown in Figure 6 is for calibrating our sensors to O_3_. The ozone monitor is connected to the gas chamber output to measure the reference ozone levels inside the gas-flow chamber. When we test the sensors with NO_2_, we augment the setup with a 2B Technologies Model 405 nm NO_x_ Reference Monitor designed for the direct measurement of NO_2_ and NO. For gas mixtures involving both O_3_ and NO_2_, we connect both of these monitors between the gas chamber and the calibration source. 

## 3. Results

Gas testing for sensor calibration involves measuring the sensor responses continuously using gas mixtures, with input levels of O_3_ between 60 and 120 ppb, NO_2_ between 20 and 80 ppb, and three levels of relative humidity (10%, 35%, 70% RH). As the focus of this paper is on developing a response surface methodology using laboratory measurements, we considered only dry air (10% RH) in this investigation. The measurement duration of the adsorption and desorption cycles is 90 min each. During adsorption, a preset gas concentration of the mixture components is introduced into the chamber from the gas source, and during desorption, the gas is replaced by dry air. A flow rate of around 150–160 mL/min is maintained throughout the experiment. The sensors are calibrated against the precise gas levels in the chamber measured by the reference instruments. For each measurement, the impedance magnitude and phase are measured at 5 kHz (arbitrarily chosen considering the signal-to-noise ratio) using the IMC.

### 3.1. Ozone Sensor Response

Figure 7 and Figure 8 plot the magnitude and phase responses, respectively, of the measured impedance of the ozone sensor for dry air gas mixtures. The preset gas concentrations, as measured by the reference instruments, are denoted by the green and blue dashed lines for O_3_ and NO_2_, respectively. The nominal preset levels for O_3_ are 60, 90, and 120 ppb, while those for NO_2_ are 20, 50, and 80 ppb. The first three gas pulses denote O_3_ input (single gas), the next three denote NO_2_ input (single gas), and the remaining 9 pulses denote mixtures of both gases. As our O_3_ and NO_2_ sensors utilize p-type semiconductors, the sensor impedance magnitude decreases, and the phase increases upon exposure to a strong oxidant gas (see the black lines). The responses indicate sensitivity to the analyte gas even at the low levels of 20 ppb NO_2_ and 60 ppb O_3_, which are relevant to public health safety [8,9]. Ideally, the maxima of the magnitude response (Figure 7) should line up with the first peak known as baseline (response of the sensor to dry air). However, the peak amplitudes decrease from the baseline due to slow and incomplete desorption. This drift in the maxima is modeled as a second-order exponential signal (magenta line). The model is used to correct the measured response, as will be described shortly. For the phase response (Figure 8), the baseline is modeled using the minima at the beginning of each adsorption pulse. 

After correcting the small baseline differences between the response pulses, we compute the magnitude response $\left|\Delta Z\right|=Z_{bsl}-Z$ and the phase response $\Delta \phi =\phi -\phi_{bsl},$ where Z and ϕ are the measured impedance magnitude and phase, respectively, and the subscript *bsl* denotes the baseline model in Figure 7 and Figure 8. These baseline-compensated measured responses are plotted in Figure 9 and Figure 10 (black lines). Except for minute deviations due to fluctuations in the data, the corrected baseline is very close to zero, as expected.

The chemical reaction of the sensor surface with the analyte gas molecules plays a very important role in sensor performance. The active layer (i.e., the gas-specific functionalization on the CNTs), nanomaterial type (semiconducting, metallic, porous, etc.), and the gas concentration determine the response change, adsorption, and desorption rates. The response, under tightly controlled operating conditions (i.e., constant flow rate and fixed operating temperature), typically involves a monotonically saturating smooth change in resistance across the sensing layer due to the adsorption of the chemical analyte occurring at the micro-porous sensor surface. Figure 11 depicts the typical steady-state response of a chemical gas sensor [102]. 

The curve in Figure 11 shows the three phases of a measurement: baseline measurement (made with pure dry air), test gas measurement (when the chemical analyte is injected in the gas phase into the test chamber), and recovery phase (during which the sensor is again exposed to pure air to return the sensor to baseline). The second phase describes the adsorption of gas molecules on the sensor surface, and the third denotes the desorption of these charge carriers to recover the sensor surface to baseline and prepare the sensor for the next reaction. The recovery time is usually much longer than the gas response time.

The measured sensor responses in Figure 9 and Figure 10 do not have the ideal characteristics shown in Figure 11 due to the slow desorption times. Slow desorption does not fully recover the sensor to baseline, resulting in cumulative baseline drift, which may eventually lead to sensor saturation. In order to determine the steady-state features, we fit the adsorption and desorption responses of both impedance magnitude and phase to an exponential signal of the form
(1)$$\begin{array}{l} R_{a}(t)=a_{i}\left(1-\exp(-(t-t_{i})/\tau_{a})\right),\;t_{i}\leq t\leq t_{i}+T\\ R_{d}(t)=b_{i}\exp(-(t-t_{i}-T)/\tau_{d}),\;t_{i}+T\leq t<t_{i+1} \end{array}$$
where the subscripts *a* and *d* stand for adsorption or desorption, $t_{i}$ denotes the beginning and *T* the duration of the *i*-th adsorption pulse (see Figure 11), and $\tau_{a}$ and $\tau_{d}$ are the time constants (rates of decay) of adsorption and desorption, respectively. Enforcing continuity of the amplitudes at $t=t_{i}+T$, we determine the steady-state value,
(2)$$b_{i}=a_{i}\left(1-\exp(-T/\tau_{a})\right)$$

The baseline-corrected magnitude response $\left|\Delta Z\right|$ and the phase response Δϕ, plotted in Figure 9 and Figure 10, are considered for extracting the steady-state features. For each gas exposure cycle, the exponential fit in (1) is computed and plotted as the red lines in these two figures. The responses are accurately modeled by the exponential signal in (1). The response time for each gas adsorption pulse, measured as the time required for the pulse response to reach 63% of the steady state, is computed from these graphs to be between 6.4 and 23 min (see Section 3.4). However, the sensor response does not recover its baseline fully even after 90 min of exposure to dry air. From the response to pulses containing only NO_2_, it is clear that the ozone sensor responds to NO_2_ (being an oxidizing gas). However, the NO_2_ responses are smaller in magnitude and phase than the O_3_ responses, as evident upon comparison of the first three (O_3_-only) pulse responses with the next three (NO_2_-only). The mixture responses of the ozone sensor (pulses 7 to 15) are predominantly controlled by the O_3_ exposure levels, but because of cross-sensitivity to NO_2_, it is observed that for a fixed O_3_ level in the mixture, as NO_2_ is increased, the sensor response decreases.

### 3.2. Nitrogen Dioxide Sensor Response

Figure 12 and Figure 13 plot the magnitude and phase responses, respectively, of the measured impedance of the NO_2_ sensor for dry air gas mixtures. The magnitude and phase responses are corrected for baseline deviation using a second-order exponential model, which is plotted as the red lines. 

The baseline-corrected magnitude response $\left|\Delta Z\right|$ and the phase response Δϕ are plotted in Figure 14 and Figure 15, respectively, for the NO_2_ sensor. For each gas exposure cycle, the exponential fit in (1) is computed and plotted as the red lines in these two figures. The adsorption response times for the NO_2_ sensor are similar to those of the O_3_ sensor. However, the desorption is much slower compared to the O_3_ sensor, especially for the magnitude. 

### 3.3. Maximum Response Change

It is clear from the analysis in Section 3.1 and Section 3.2 that the sensor array responds to both O_3_ and NO_2_ analytes in a nonselective manner. In general, it is observed that the magnitude change is much smaller than the phase change for the NO_2_ sensor. In contrast, both magnitude and phase changes are equally prominent for the O_3_ sensor (see Figure 9 and Figure 10). These differences are quantified in Table 1 as a percentage relative to the pulse baseline (i.e., the baseline model value at the beginning of the adsorption). The maximum magnitude and phase changes are indicated for the target gas, as well as the interfering gas, along with the corresponding adsorption pulse numbers. It is observed that cross-sensitivity is significant in the array. 

### 3.4. Response Time

The response time for gas adsorption, measured as the time required for the pulse response to reach 63% of the steady state (or one time constant), is computed from the baseline-corrected fitted response graphs shown in Figure 9 and Figure 10 for the O_3_ sensor, and Figure 14 and Figure 15 for the NO_2_ sensor. The fitted response is an exponential, and its time constant can be analytically determined. The response time of each gas pulse in the baseline-corrected model is listed in Table 2 for the impedance magnitude and phase of both O_3_ and NO_2_ sensors. The average response time is about 11 min for the O_3_ magnitude and 14 min for the NO_2_ magnitude. The average response time of the impedance phase is calculated as 14 min for the O_3_ sensor and 18 min for the NO_2_ sensor. The response at low concentrations is slower than at higher concentrations. This is clearly evident in Table 2. As an example, high-concentration gas pulses 11 to 15 for the ozone sensor phase depict response times between 3 and 6 min, compared to between 18 and 26 min for the low-concentration pulses 4 to 6. As the majority of the pulses did not exhibit complete desorption within the observed pulse duration of 90 min, the desorption times have not been calculated.

### 3.5. Response Surface Fitting

We now discuss the multivariate response surface methodology to determine a mathematical relationship between the gas sensor response and its concentration levels. We recall that the investigated gas sensor array produces impedance magnitude and phase responses when exposed to specific levels of O_3_ and NO_2_ fed into the gas chamber. The nominal preset levels for O_3_ are 60, 90, and 120 ppb, while those for NO_2_ are 20, 50, and 80 ppb. The baseline response is activated by zero air (i.e., 0 ppb gas level). Thus, there are two factors (or gases), each with four levels (including 0 ppb), which yields a 4^2^ or 16-run full-factorial experiment. The baseline drift-corrected magnitude and phase responses of the O_3_ sensor are depicted in Figure 9 and Figure 10, respectively, while those for the NO_2_ sensor are shown in Figure 14 and Figure 15. In each figure, the concentration levels of the component gases are presented in the reference gas profiles. As an example, in gas pulse number 3, the nominal reference O_3_ level is 120 ppb and the NO_2_ level is 0 ppb, while in gas mixture pulse number 8, these levels are 120 ppb and 40 ppb, respectively. Note that the actual measured gas levels from the depicted gas profiles are used in the response surface fitting instead of the nominal levels. For each (O_3_, NO_2_) gas pulse, the steady state values of the magnitude and phase responses were recorded from the curve-fitted models. These steady-state values comprise the variables to which the response surface is fitted in the (O_3_, NO_2_) plane. It is noted that the first NO_2_ pulse (number 4 in the baseline-corrected responses—see Figure 9, Figure 10, Figure 14 and Figure 15) has poor data quality, showing a much larger response change in desorption than in adsorption. This pulse is discarded as an outlier in the response surface estimation.

The response surface describes an optimized mathematical representation of the magnitude or phase response in terms of gas concentration levels. The relationship between the steady-state response levels and the gas levels in our full-factorial experiment is modeled by a second-order polynomial as
(3)$$R_{mn}=p_{mn}+q_{mn}x+r_{mn}y+u_{mn}x^{2}+w_{mn}xy+v_{mn}y^{2}$$
where m=1, 2 indicates the sensor type (1 = O_3_ sensor, 2 = NO_2_ sensor), and n=1, 2 denotes the response feature (1 = magnitude response and 2 = phase response), respectively. Thus, $R_{11}=\left|\Delta Z_{1}\right|,\;R_{12}=\Delta \Phi_{1}$ signify magnitude and phase responses at steady state for the O_3_ sensor, while $R_{21}=\left|\Delta Z_{2}\right|,\;R_{22}=\Delta \Phi_{2}$ signify those for the NO_2_ sensor. The variables *x* and *y* denote the measured concentrations (in ppb) of ozone and NO_2_, respectively. The units for $\left|\Delta Z\right|\;\mathrm{and}\;\Delta \Phi$ are MΩ and degrees, respectively. The model coefficients in Equation (3) are determined by minimization of the fitting error using the Levenberg Marquardt nonlinear optimization method in *Matlab* programming and numeric computing platform (https://www.mathworks.com/products/matlab.html (accessed on 15 June 2022). The fitted response surface for the O_3_ sensor is displayed in Figure 16 for the impedance magnitude and in Figure 17 for the impedance phase. Table 3 contains the model coefficients and the goodness-of-fit parameters. The RMS prediction error (standard deviation of the residuals) is only 19.9 kΩ with R^2^ = 0.96 for the magnitude, while the RMS error is 0.45 degrees with R^2^ = 0.94 for the phase. The residuals (not plotted) appear randomly scattered around zero, indicating that the model describes the data well without any significant bias. The higher values of R^2^ in each response surface fit reinforce that most of the variability in the data is captured by the model. 

The fitted response surface of the NO_2_ sensor is displayed in Figure 18 for the impedance magnitude and Figure 19 for the impedance phase, respectively. Table 4 contains the model coefficients and the goodness-of-fit parameters. Although R^2^ values are lower than those for the O_3_ sensor, the small RMS error (4.7 kΩ in magnitude and 0.1 degrees in phase) indicates a good fit.

### 3.6. Sensor Performance Evaluation

#### 3.6.1. Sensitivity

The response surface models for the array sensor responses to gas mixtures of O_3_ and NO_2_ allow us to analyze the sensor performance in terms of sensitivity and selectivity within the range of 60–120 ppb for O_3_ and 20–80 ppb for NO_2_. Below this range, adequate measured data do not exist to make reasonable conclusions from the model. Sensitivity is calculated by differentiating the fitted response R(x,y) in Equation (3) with respect to the ozone concentration x or NO_2_ concentration y. Using the notation introduced in Equation (3), i.e., subscript *m* to denote the sensor and *n* to signify the response magnitude or phase feature, the impedance magnitude and phase sensitivities are calculated using Equations (4) and (5), respectively.
(4)$$S_{z}=\left[\begin{array}{cc} \frac{\partial R_{11}}{\partial x} & \frac{\partial R_{11}}{\partial y}\\ \frac{\partial R_{21}}{\partial x} & \frac{\partial R_{21}}{\partial y} \end{array}\right]≜\left[\begin{array}{cc} s_{1x} & s_{1y}\\ s_{2x} & s_{2y} \end{array}\right]$$
(5)$$S_{\phi }=\left[\begin{array}{cc} \frac{\partial R_{12}}{\partial x} & \frac{\partial R_{12}}{\partial y}\\ \frac{\partial R_{22}}{\partial x} & \frac{\partial R_{22}}{\partial y} \end{array}\right]≜\left[\begin{array}{cc} \psi_{1x} & \psi_{1y}\\ \psi_{2x} & \psi_{2y} \end{array}\right]$$

The sensitivity couplets $\left(s_{mx},\;\psi_{mx}\right)$ and $\left(s_{my},\;\psi_{my}\right)$ denote magnitude and phase sensitivities of the sensor *m* to ozone and NO_2_, respectively. These sensitivities are easily calculated by partial differentiation with respect to *x* or *y* of the polynomials in Equation (3). Due to the nonlinearity of the sensor response, each sensitivity depends on both component analyte concentrations in the gas mixture. 

Figure 20 displays the calculated sensitivity $s_{1x}$ for the impedance magnitude, and $\psi_{1x}$ for the impedance phase, of the ozone sensor. The ozone sensitivity is plotted at a few fixed levels of NO_2_. The peak sensitivity occurs at 60 ppb of ozone, and the sensitivity decreases with further increase in ozone exposure. The magnitude response exhibits a marginal reduction in ozone sensitivity with an increase in the NO_2_ level from 20 to 80 ppb, indicating that cross-sensitivity to NO_2_ is present. The phase response seems remarkably immune to this cross-sensitivity. Next, we examine the cross-sensitivity of the ozone sensor to NO_2_, represented by $s_{1y}$ for the impedance magnitude and $\psi_{1y}$ for the impedance phase. These two cross-sensitivities are plotted in Figure 21. The cross-sensitivity becomes negative, degrading the response of the ozone sensor to its target gas. This is illustrated from the response of the ozone sensor to the gas mixture given by
(6)$$\begin{array}{l} R_{11}=s_{1x}c_{x}+s_{1y}c_{y}\\ R_{12}=\psi_{1x}c_{x}+\psi_{1y}c_{y} \end{array}$$
where $c_{x}$ and $c_{y}$ denote O_3_ and NO_2_ concentrations, respectively. It is evident that negative $s_{1y}$ and $\psi_{1y}$ subtract from the response to the target gas and impact the selectivity.

Although sensitivity and cross-sensitivity have been computed from the model in Equation (3) for the NO_2_ sensor magnitude and phase responses, they are not plotted for brevity. It is observed that the cross-sensitivity to O_3_ is significantly high, indicating that the NO_2_ sensor has poor selectivity. The selectivity of both sensors will be investigated next. 

#### 3.6.2. Selectivity

Selectivity is computed using the definitions introduced in [114]. The selectivity of a sensor is defined in general as
(7)$$\kappa =\frac{\mathrm{sensor}\;\mathrm{response}\;\mathrm{to}\;\mathrm{target}\;\mathrm{gas}}{\mathrm{sensor}\;\mathrm{response}\;\mathrm{to}\;\mathrm{gas}\;\mathrm{mixture}}$$

For each of the sensors in the array, we measured both magnitude and phase responses and determined the response surfaces for each response. Selectivity may be expressed in terms of the sensitivities of each response feature and the corresponding component analyte level in the gas mixture. Thus,
(8)$$\kappa_{m}=\frac{\left(\left|s_{mp}\cos\psi_{mp}\right|+\left|s_{mp}\sin\psi_{mp}\right|\right)c_{p}}{\left(\left|s_{mp}\cos\psi_{mp}\right|+\left|s_{mp}\sin\psi_{mp}\right|\right)c_{p}+\left(\left|s_{mq}\cos\psi_{mq}\right|+\left|s_{mq}\sin\psi_{mq}\right|\right)c_{q}}$$
where the subscript *m* denotes the sensor (1 for O_3_ and 2 for NO_2_), $c_{p}$ denotes concentration of the target gas, and $c_{q}$ denotes concentration of the interfering gas. Thus, for the ozone sensor, p=x represents the ozone target and q=y denotes the interfering NO_2_ gas, and vice versa for the NO_2_ sensor. The trigonometric functions evaluate the real and imaginary parts of the complex response of the form $R_{mp}=s_{mp}\exp(j\psi_{mp})c_{p}.$

The second term in the denominator in Equation (8) indicates the cross-sensitivity component due to the interfering gas with concentration $c_{q}.$ The maximum selectivity of unity occurs when the cross-sensitivity is zero. At the other extreme, when the sensor does not respond to the target gas, the selectivity is zero. Thus, the selectivity ranges between 0 and 1. In practice, selectivity around 0.9 is considered a good performance metric. Next, we discuss the results on the selectivity of both sensors. 

The line plot of selectivity of the O_3_ sensor, computed from Equation (8), is depicted in Figure 22 as a function of O_3_ concentration at a few constant NO_2_ levels. The selectivity to ozone is highest for NO_2_ levels between 20 and 30 ppb and degrades rapidly at higher levels, indicating significant cross-sensitivity to NO_2_ beyond 30 ppb. 

It is emphasized that although many gases are present in the environment, NO_2_ is the strongest interferent to O_3_, and vice versa. We have earlier reported [97] on evaluating the cross-sensitivity of the ODA-functionalized ozone sensor to various environmental interferents, including NO_2_, NH_3_, CH_4_ and CO_2_, and found that except NO_2_ none of the other gases are cross-sensitive to O_3_ detection. This has also been concluded by commercial sensor manufacturers [35,36,37,38], one of whom uses limited-lifetime membrane filters in their O_3_ electrochemical sensors to absorb the NO_2_ interferent [38]. However, this filter saturates within weeks and thus needs frequent replacement, whereas CNTs can be designed to be selective over long periods of exposure.

Next, we examine the selectivity of the NO_2_ sensor to its target gas, computed using (8). Figure 23 displays the line graph of the selectivity for the NO_2_ sensor as a function of the sample analyte component levels in the gas mixture. The selectivity to NO_2_ is highest at NO_2_ = 80 ppb and O_3_ at 60 ppb or less. The selectivity is significantly degraded at lower levels of NO_2_. The maximum selectivity of NO_2_ sensor is only 0.65, indicating that cross-sensitivity to O_3_ drastically affects the performance of the NO_2_ sensor. In contrast, it is recalled that NO_2_ levels less than 30 ppb have minimal interference in the selectivity response of the O_3_ sensor (see Figure 22). 

## 4. Discussion

The sensors presented in this paper utilize functionalized single-walled CNTs, deposited as thin films on printed interdigitated electrodes, to detect O_3_ and NO_2_ at environmentally relevant low concentrations below 120 ppb. Our approach exploits the unique electronic properties of CNTs to modulate their sensitivity by using chemical functionalization tailored for the adsorption of specific molecules. The innovation in this work lies in improving the sensitivity to the target gas in a gas mixture containing oxidizing interferents, such as O_3_ and NO_2_ present in the measurement environment. We characterized in the laboratory an array comprising two gas sensors, one designed to be more specific to O_3_ and the other to NO_2_. Both O_3_ and NO_2_ sensors exhibit cross-sensitivity and produce nonselective responses. *We quantified the selectivity to the target analyte, as well as the sensitivity and cross-sensitivity of each sensor, using response surface methodology applied to the measured impedance magnitude and phase steady-state responses when exposed to gas mixtures within the range of 60–120 ppb for O*_3_
*and 20–80 ppb for NO*_2_. 

Some limitations of the devices were observed, most notably, slow (incomplete) recovery to baseline (i.e., slow desorption), which can potentially result in long-term baseline drift. However, it is gratifying that the sensors did not saturate even over extended periods (nearly 48 h) of gas exposure during laboratory testing. However, sensor drift, saturation, and stability need to be investigated by measuring the sensor responses over longer intervals spanning several months. The effects of meteorological parameters, such as humidity and temperature, also need to be studied and accounted for. We anticipate the investigation of these aspects of sensor performance after the optimization of the sensor response times and desorption. In order to sense ambient environments, the calibrated sensors will eventually be fielded in outdoor locations, and a machine learning pattern recognition algorithm will be utilized with the response surface formulation to determine the ambient O_3_ and NO_2_ concentrations from the measured magnitude and phase responses. These ambient concentrations will be validated against the measurements from collocated precision reference-grade instruments and/or the regulatory EPA monitors in the vicinity of the sensors. 

### Performance Comparison with Prior Research

Carbon-based nanostructured materials, such as single-walled and multi-walled carbon nanotubes, as well as their functionalization with nanoparticles comprising metals or metal oxides, have been used by researchers to improve the performance of gas sensors [110,112,115,116,117,118,119,120]. Table 5 lists the performance of a few representative O_3_ and NO_2_ sensors studied in prior research efforts, and evaluates their comparison with the results presented in this paper. It is emphasized that the comparison should be interpreted in a qualitative sense, since the properties of a CNT gas sensor depend not only on the materials used but also on how the materials are integrated and processed into sensor films for deposition on the electrodes, as well as the substrates and metallization used to design the electrodes. Important parameters to consider include the type of response measured, gas concentration, sensor operating temperature, response change or sensitivity, and response time. Although response times can be faster in some sensor implementations than others, as seen in Table 5, the sensor recovery or desorption time is usually of the order of several minutes to an hour or more, especially for room-temperature sensors (cf. [112]) and low-temperature sensors (this work). Response times generally decrease with increasing concentrations, but higher concentrations tend to saturate the sensors even at the relatively high operational temperatures [117,118,119] employed for desorption of the gas molecules. The ODA-functionalized ozone sensors presented herein have a relatively fast response of 6 to 14 min in comparison with the hybrid SnO_2_ decorated SWNT sensor in [117] but much higher response change than all the prior results in Table 5 for O_3_ detection. The amide-ZnO functionalized SWNT sensor for NO_2_ presented in this work is predominantly chemi-resistive (i.e., almost zero phase, see Figure 15), and did not perform as well as expected. 

Slow sensor recovery and potential saturation are important barriers to be overcome for the successful deployment of CNT-based sensors in ambient air quality measurements. Strong binding of the analyte molecules contributes to slow desorption in environmental sensors. At the required low concentration levels of the analyte to be detected, the charge carriers embedded on the sensor surface by the adsorption of gas molecules become tightly bound to the surface, and it takes a strong external thermal or electrical stimulus (e.g., heating the sensor or UV illumination) for rapid desorption of the attached gas molecules. In the future, we plan to conduct detailed temperature cycling tests of SWNT films, including rapid temperature modulation using random thermal waveforms, to facilitate fast sensor recovery. In order not to damage the sensors, in the initial study reported herein, sensor heating was restricted to a conservative 75 °C to 80 °C range. Based on the literature [115,116,117,118,119,120], and our own TGA analysis for both O_3_-senstive and NO_2_-sensitive CNT films (cf. Figure 1b and Figure 2a), it seems feasible to obtain smaller desorption or recovery times by heating the sensor between 150 °C and 200 °C. This is still substantially lower heating than MOS sensors, which require operational temperature in the 350–450 °C range. 

## Figures and Tables

**Figure 1 sensors-23-08447-f001:**
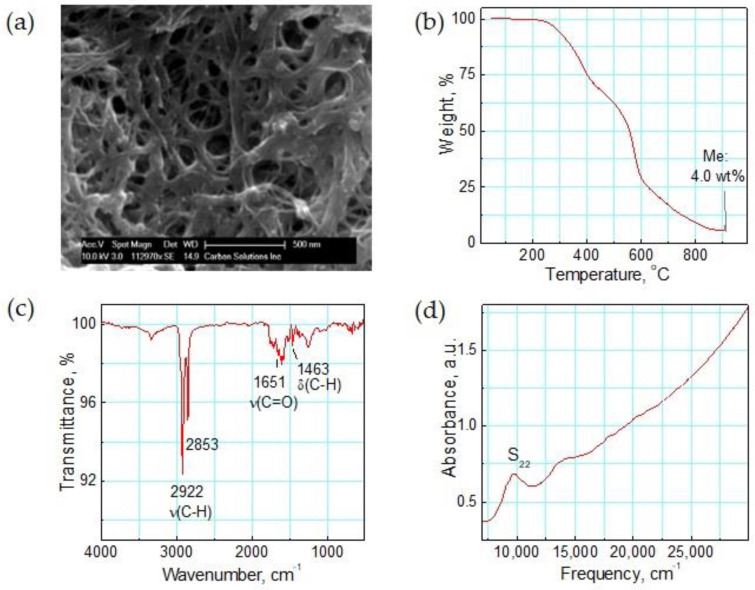
Characterization data for ODA-functionalized SWNTs: (**a**) SEM image showing the porous nanotube networks, (**b**) TGA in air, and (**c**) mid-IR and (**d**) near-IR spectroscopy.

**Figure 2 sensors-23-08447-f002:**
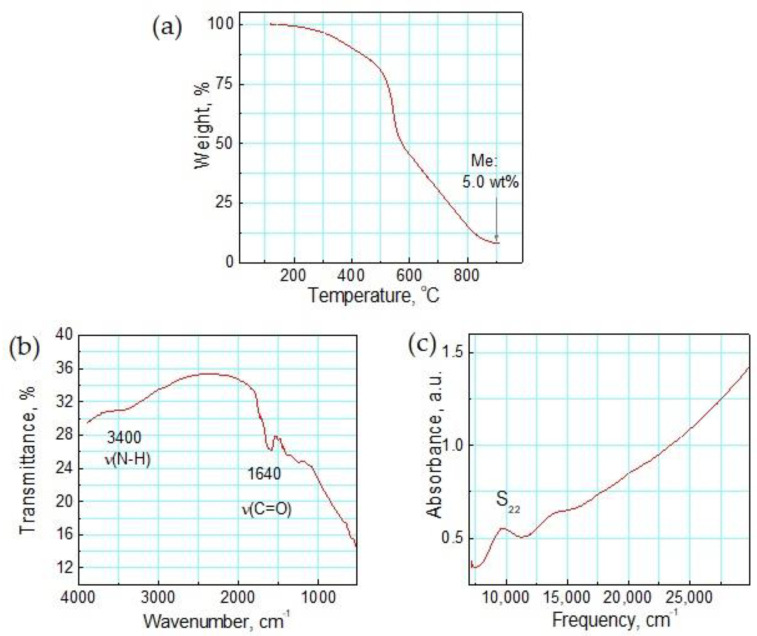
Characterization data for amide-functionalized, ZnO-decorated SWNTs: (**a**) TGA in air, and (**b**) mid-IR and (**c**) near-IR spectroscopy.

**Figure 3 sensors-23-08447-f003:**
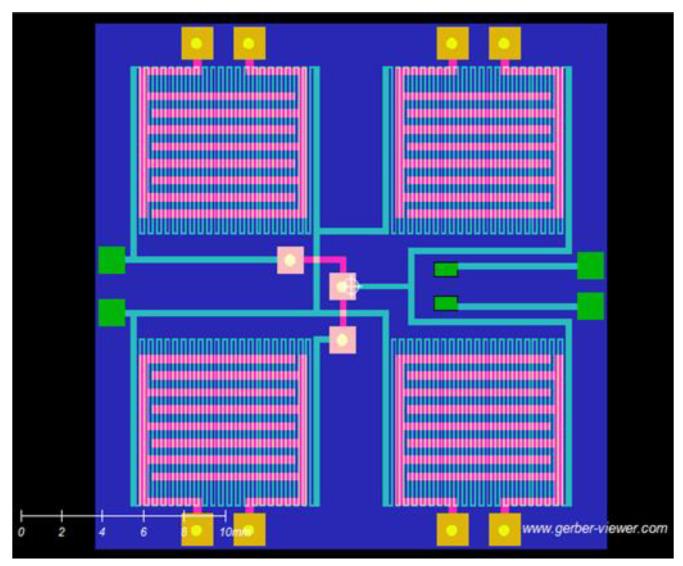
PCB layout of the sensor array design. The transparent view is a projection of the meandered line heater from the top and the four pairs of IDEs from the bottom. Each sensor is accompanied by its own heater.

**Figure 4 sensors-23-08447-f004:**
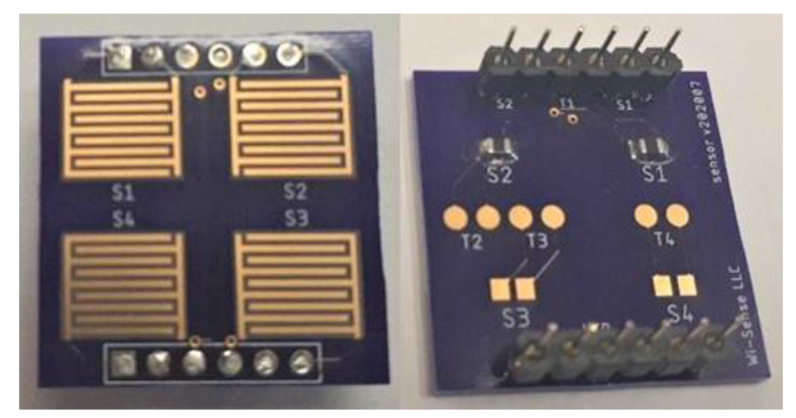
Fabricated sensor array board. Top view (**left**), bottom view (**right**).

**Figure 5 sensors-23-08447-f005:**
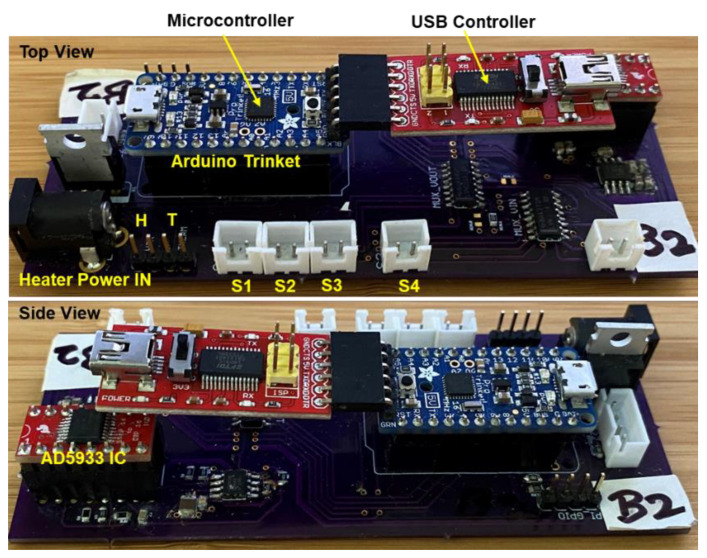
Electronics printed circuit board for the impedance measurement circuit (IMC). The sensors are connected to terminals S1–S4. The heater is connected to terminal H, and the thermistor temperature is monitored using terminal T.

**Figure 6 sensors-23-08447-f006:**
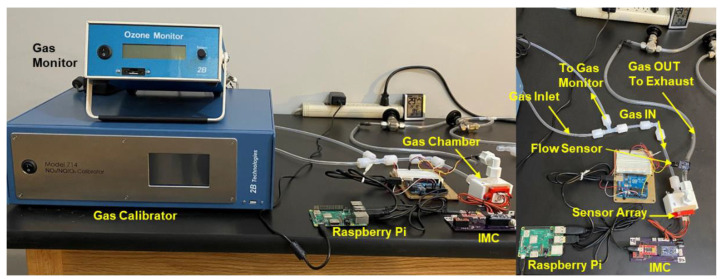
Controlled gas flow calibration measurement setup using the IMC to characterize the electrical response and a Raspberry Pi to record and communicate the data.

**Figure 7 sensors-23-08447-f007:**
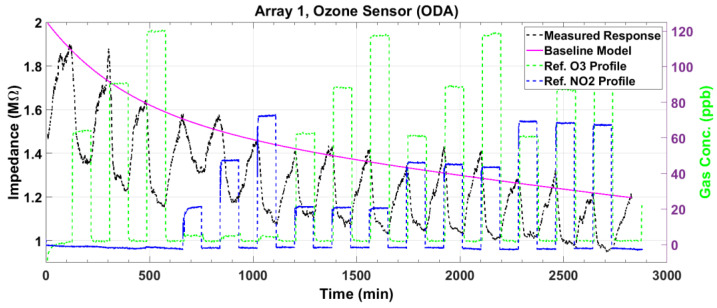
The measured impedance magnitude response (black line) of the ozone sensor to gas mixtures at various concentrations of O_3_ and NO_2_. The deviation from the baseline is modeled as a second-order exponential (magenta line) to correct the response.

**Figure 8 sensors-23-08447-f008:**
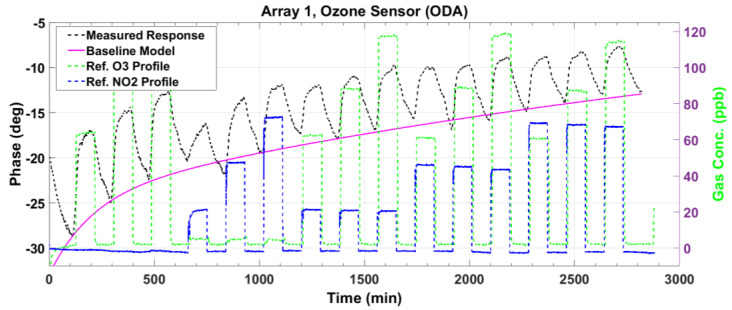
The measured phase response (black line) of the ozone sensor to gas mixtures at various concentrations of O_3_ and NO_2_. The deviation from the baseline is modeled as a second-order exponential (magenta line) to correct the response.

**Figure 9 sensors-23-08447-f009:**
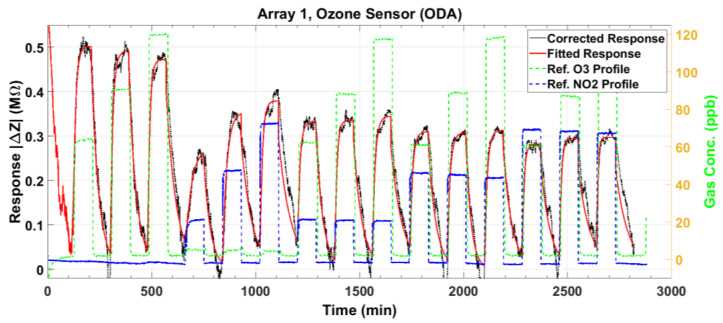
The baseline-corrected measured and modeled (fitted) impedance magnitude response of the ozone sensor at various concentrations of O_3_ and NO_2_.

**Figure 10 sensors-23-08447-f010:**
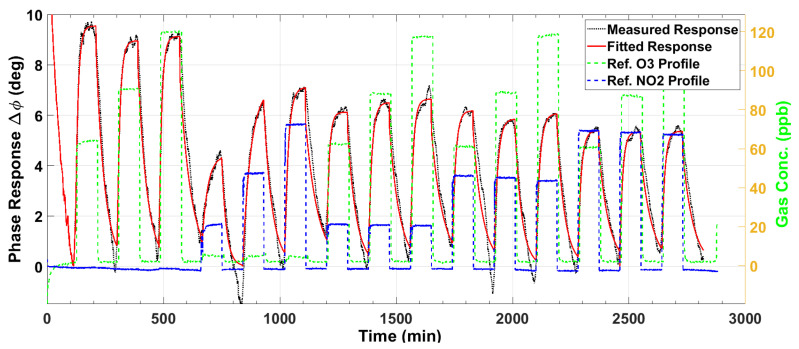
The baseline-corrected measured and modeled (fitted) impedance phase response of the ozone sensor at various concentrations of O_3_ and NO_2_.

**Figure 11 sensors-23-08447-f011:**
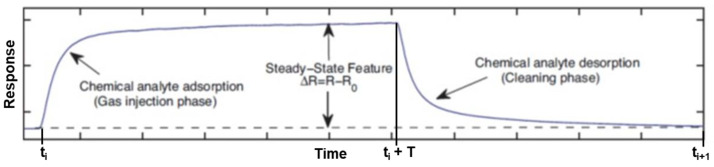
Typical response of a gas sensor describing steady-state adsorption and desorption phases [102] (reproduced with permission from Elsevier, Amsterdam, The Netherlands).

**Figure 12 sensors-23-08447-f012:**
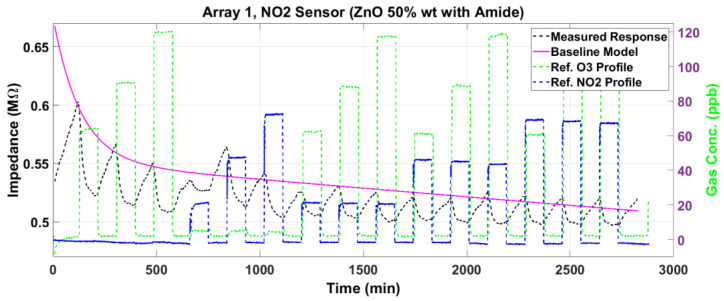
The measured impedance magnitude response (black line) of the NO_2_ sensor to gas mixtures at various concentrations of O_3_ and NO_2_. The deviation from the baseline is modeled as a second-order exponential (magenta line) to correct the response.

**Figure 13 sensors-23-08447-f013:**
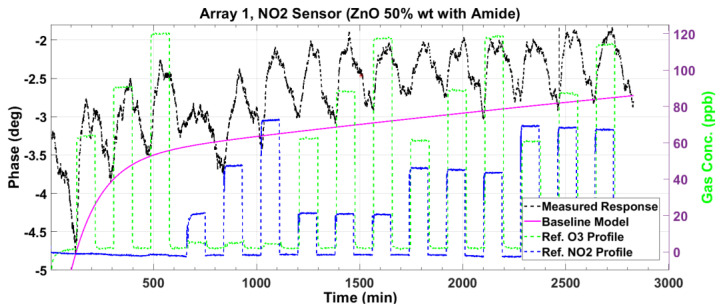
The measured impedance phase response (black line) of the NO_2_ sensor to gas mixtures at various concentrations of O_3_ and NO_2_. The deviation from the baseline is modeled as a second-order exponential (magenta line) to correct the response.

**Figure 14 sensors-23-08447-f014:**
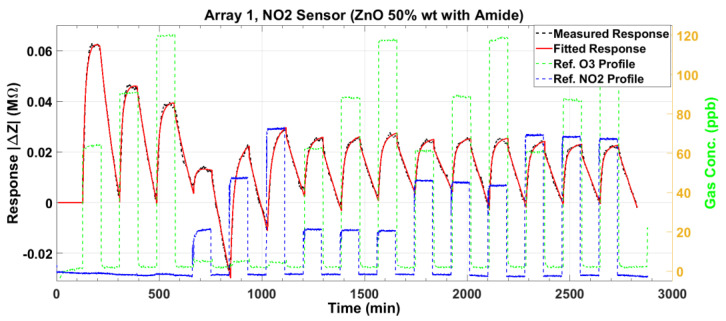
The baseline-corrected measured and modeled (fitted) impedance magnitude response of the NO_2_ sensor at various concentrations of O_3_ and NO_2_.

**Figure 15 sensors-23-08447-f015:**
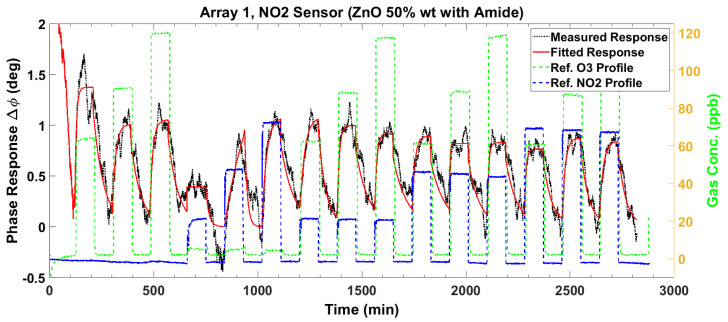
The baseline-corrected measured and modeled (fitted) impedance phase response of the NO_2_ sensor at various concentrations of O_3_ and NO_2_.

**Figure 16 sensors-23-08447-f016:**
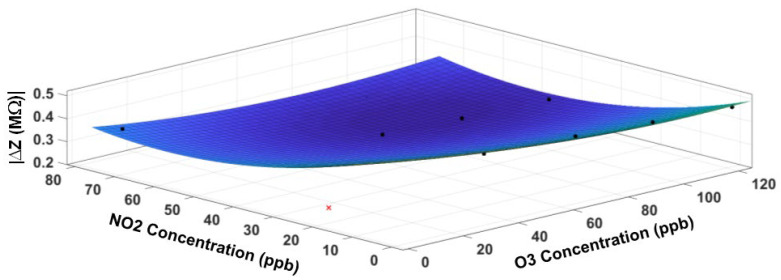
The response surface fit for the impedance magnitude of the O_3_ sensor. The cross symbol indicates an outlier in the data, which is excluded from the model. The circle symbol indicates data points (not all data points visible).

**Figure 17 sensors-23-08447-f017:**
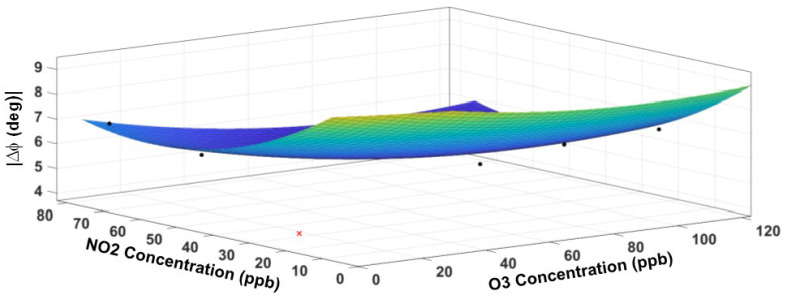
The response surface fit for the impedance phase of the O_3_ sensor. The cross symbol indicates an outlier in the data, which is excluded from the model. The circle symbol indicates data points (not all data points visible).

**Figure 18 sensors-23-08447-f018:**
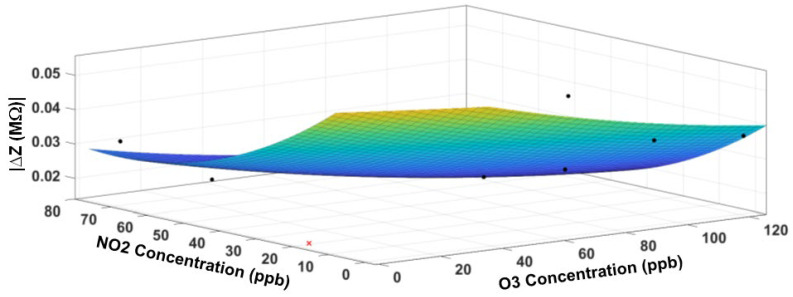
The response surface fit for the impedance magnitude of the NO_2_ sensor. The cross symbol indicates an outlier in the data, which is excluded from the model. The circle symbol indicates data points (not all data points visible).

**Figure 19 sensors-23-08447-f019:**
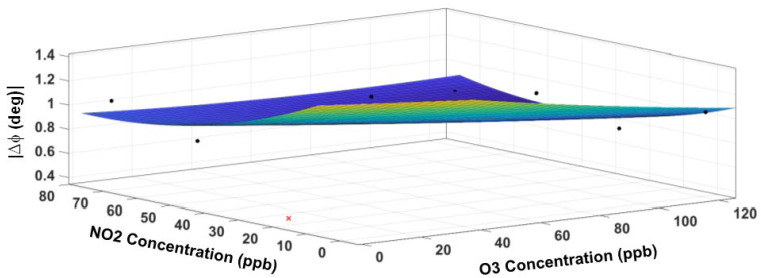
The response surface fit for the impedance phase of the NO_2_ sensor. The cross symbol indicates an outlier in the data, which is excluded from the model. The circle symbol indicates data points (not all data points visible).

**Figure 20 sensors-23-08447-f020:**
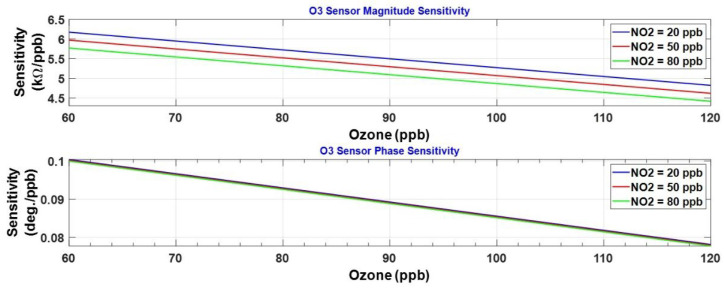
Impedance magnitude and phase sensitivities of the O_3_ sensor.

**Figure 21 sensors-23-08447-f021:**
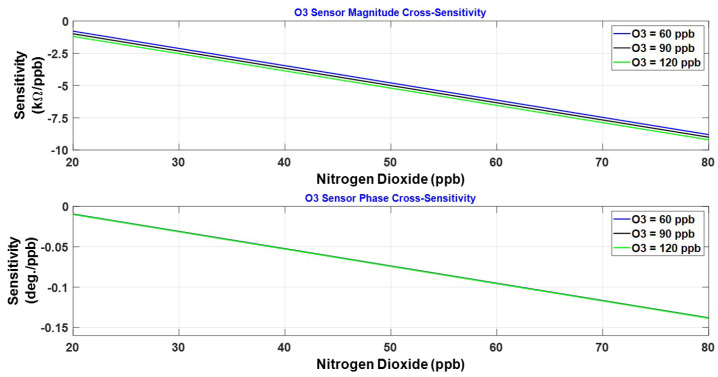
Impedance magnitude and phase cross-sensitivities of the O_3_ sensor.

**Figure 22 sensors-23-08447-f022:**
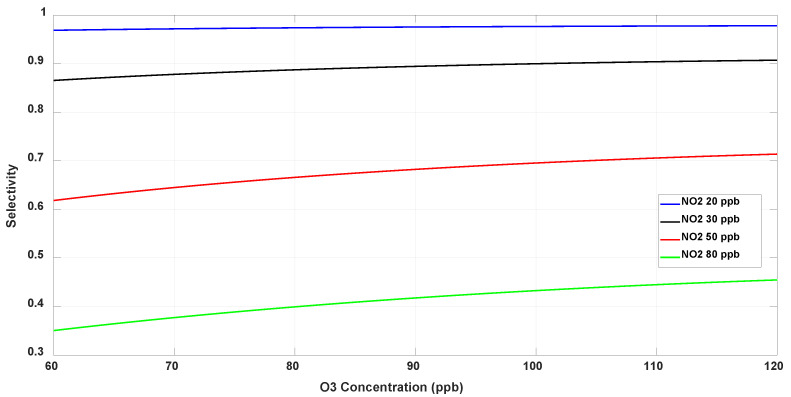
Selectivity of the O_3_ sensor at fixed NO_2_ levels.

**Figure 23 sensors-23-08447-f023:**
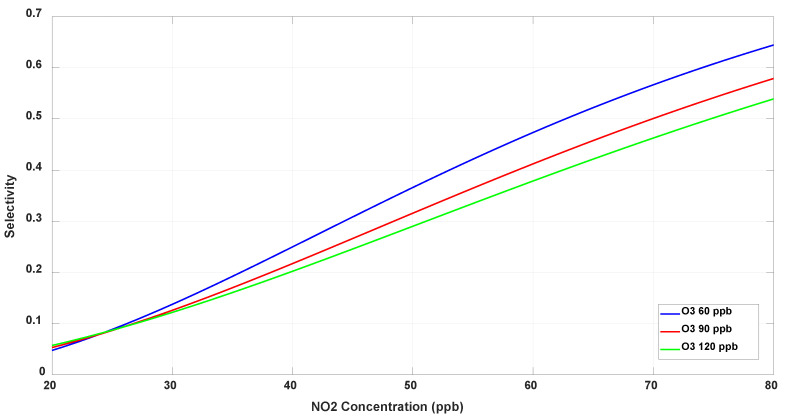
Selectivity of the NO_2_ sensor at fixed O_3_ levels.

**Table 1 sensors-23-08447-t001:** Comparison of maximum response change between the O_3_ and NO_2_ sensors.

Sensor	Gas/Pulse No.	Max |ΔZ|/Z_0_	Max |Δφ|/φ_0_	Comment
O_3_	O_3_/3	26.8%	32.9%	Target gas
O_3_	NO_2_/6	26.0%	36.4%	Interfering gas
NO_2_	NO_2_/6	5.3%	30.9%	Target gas
NO_2_	O_3_/3	10.3%	28.6%	Interfering gas

**Table 2 sensors-23-08447-t002:** Adsorption response time (in minutes) of the O_3_ and NO_2_ sensors. The numbers in the first row identify the gas pulses injected into the gas flow chamber.

Feature	1	2	3	4	5	6	7	8	9	10	11	12	13	14	15	Avg
O_3_ Mag.	6.38	13.47	10.33	10.85	21.24	13.70	8.40	9.48	14.83	11.79	7.99	9.06	7.44	7.41	11.15	10.90
O_3_ Phase	13.64	15.60	13.56	18.24	23.82	21.54	25.52	14.25	17.10	8.33	3.02	5.69	2.77	6.56	4.95	12.97
NO_2_ Mag.	18.42	13.80	12.68	15.87	24.55	25.05	12.06	14.18	12.98	14.16	10.08	7.81	11.40	8.74	8.40	14.01
NO_2_ Phase	14.73	15.64	18.66	21.84	32.19	19.47	16.89	12.79	17.55	15.43	14.42	13.8	15.18	14.4	19.89	17.52

**Table 3 sensors-23-08447-t003:** Fitted model coefficients and goodness-of-fit for the O_3_ sensor responses.

Response	p	q	r	u	w	v	R^2^	RMSE
Magnitude	0.581	−2.288 × 10^−3^	−7.663 × 10^−3^	1.128 × 10^−5^	6.752 × 10^−6^	6.680 × 10^−5^	0.9568	0.0199
Phase	10.34	−3.379 × 10^−2^	−0.1227	1.851 × 10^−4^	6.799 × 10^−6^	1.072 × 10^−3^	0.9376	0.4532

**Table 4 sensors-23-08447-t004:** Fitted model coefficients and goodness-of-fit for the NO_2_ sensor responses.

Response	p	q	r	u	w	v	R^2^	RMSE
Magnitude	0.06259	−3.057 × 10^−4^	−9.963 × 10^−4^	8.364 × 10^−7^	1.496 × 10^−6^	7.328 × 10^−6^	0.8849	0.0047
Phase	1.511	−4.888 × 10^−3^	−1.565 × 10^−2^	8.616 × 10^−6^	4.364 × 10^−5^	1.093 × 10^−4^	0.7178	0.1050

**Table 5 sensors-23-08447-t005:** Representative performance comparison with prior results.

Materials	Gas	Response	Concentration (ppb)	Temperature (°C)	Response Change	Response Time (min)	Reference
ODA-SWNTs ^1^	O_3_	abs (impedance)	60	75	26.8%	6.4	This work
ODA-SWNTs	O_3_	impedance phase	60	75	32.9%	13.6	This work
Pd-decorated MWNTs ^2^	O_3_	resistance	100	120	1.47	1	[116]
Hybrid SnO_2_/SWNTs	O_3_	resistance	174	300	10.5	40	[117]
Thermally treated SWNTs	O_3_	resistance	1000	350	14.7	1.7	[118]
ZnO NPs ^3^/amide-SWNTs	NO_2_	abs (impedance)	70	75	5.3%	25.1	This work
ZnO NPs/amide-SWNTs	NO_2_	impedance phase	70	75	30.9%	19.5	This work
Au NPs/SWNTs	NO_2_	resistance	12,000	240	10%/ppm	0.5	[119]
ZnO NPs/SWNTs	NO_2_	resistance	1000	25	70%	3.3	[112]
PECVD ^4^ of CNTs on Si_3_N_4_/Si substrates	NO_2_	resistance	100	165	3.3	65	[120]

^1^ Single-walled carbon nanotubes, ^2^ Multi-walled carbon nanotubes, ^3^ Nanoparticles, ^4^ Plasma-enhanced chemical vapor deposition.

## Data Availability

Not applicable.

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
