# Peer review of "Response Surface Modeling of the Steady-State Impedance Responses of Gas Sensor Arrays Comprising Functionalized Carbon Nanotubes to Detect Ozone and Nitrogen Dioxide"

_sensors, 2023, doi:10.3390/s23208447_

Round 1

Reviewer 1 Report

The paper reports the fabrication of gas sensor arrays and their sensing performance upon exposure to O3 or NO2 gas species. The material system is based on the CNT with surface modification. The sensitivity and selectivity of the sensing performance are high. The practical application is well demonstrated. I think this paper is suitable for publication in Sensors.

1. The surface modificaion of CNTs should be characterized with more observation methods.

2. How can the stability of the sensor be improved?

3. Figures 15-20 have display problem. The authors can correct them.

4. The performance can be compared with previously reported data.

Reviewer 2 Report

Reviewer's comments:

1. The inference of research data in the sensitivity of sensors should be exhibited in the “Abstract” section.

2. For confirming the responsivity, the experiment data of adsorption time and desorption time in average in the measured periods both in O3 array sensor and in NO2 array sensor should be precisely presented in the research results also.

3. It is suggested that the citation of literatures in 2023 should be included in “References” section and the performance such as sensitivity should be compared with other reported papers. 

Reviewer 3 Report

The title does not correspond to the content of the work and is not very informative. The publication does not apply to all environmental gases, but only O3 and NO2. It would be good to add that carbon nanotubes were used in the sensors.

The introduction is overall well written. In lines 68 - 73 it would be good to mention SAW sensors. Some sensors are described by acronyms and some are not. Why?

L. 123. What are ppm hours?

L. 143. Instead of mm3 should be mm.

L. 340 – 344. The results of the work are of little practical importance. The authors studied a very simple layout. In fact, much more gases are present in the environment. The RH of the atmosphere is rarely 10 %. The adsorption and desorption time of 90 min of analytes is too long during sensors application.

Figures 15 - 20 are unclear. There is no red cross in Fig.15. Figures 15 and 16 seem exactly the same. The difference between the two is only in the description in the form of impedance magnitude and phase. And a surface for adsorption should be given for O3. The similar remarks are also valid for Fig. 17, 18 and 19.

Figure 20 is not very informative. Selectivity is much better illustrated by Figures 21 and 22.

The authors announce that in the future they intend test their sensors at 120 -150 oC (l. 635-636). According to the information in the lines 430 - 432, however, this is not possible.
